# Differentially and Integrally Attentive Convolutional-based Photoplethysmography Signal Quality Classification

## Abstract

Photoplethysmography (PPG) is a non-intrusive and cost-effective optical technology that detects changes in blood volume within tissues, providing insights into the body's physiological dynamics over time. By analyzing PPG data as a time series, valuable information about cardiovascular health and other physiological parameters such as Heart Rate Variability (HRV), Peripheral Oxygen Saturation (SpO2), and sleep status can be estimated. With the ever increasing user adoption of wearable devices like smartwatches, Health Monitoring Applications (HMA) are gaining popularity due to their ability to track various health metrics, including sleep patterns, heart rate, and activity tracking, by making use of PPG sensors to monitor different aspects of an individual's health and wellness. However, reliable health indicators require high-quality PPG signals, which are often contaminated with noise and artifacts caused by movement when using wearables. Hence, Signal Quality Assessment (SQA) is crucial in determining the trustworthiness of PPG data for HMA applications. We present a new PPG SQA approach, leveraging recent advancements in differential and integral attention-based strategies coupled with a two-stage procedure for promptly discarding highly anomalous segments, as a means of enhancing the performance of Convolutional Neural Network (CNN)-based SQA classifiers, balancing storage size and classifier accuracy in resulting models of increased robustness across PPG signals from different devices. Our methods are capable of achieving F1-scores between 0.9194 and 0.9865 across four expert-annotated datasets from different wearable devices.

## 1 Introduction

The growing influence and widespread acceptance of continuous health monitoring on the market of smart wearable devices, ranging from fitness bands to smartwatches and smart rings, is remarkable. The global increase in wearable usage is evident, with projections indicating over 1 billion wearables worldwide by 2022 and annual expenditures exceeding $80 billion (Cisco, 2019; Gartner, 2021). Such devices allow customization and integration of diverse sensor types, communication units, and remote computing resources, offering holistic health solutions to users. In this line, mobile applications embedded in such devices are increasingly utilized to estimate several physiological factors from users, such as Heart Rate Variability (HRV), Peripheral Oxygen Saturation (SpO2), and sleep quality.

In this context, Photoplethysmography (PPG) emerges as a convenient technique, playing a significant part in wearable health monitoring systems, offering potentially valuable insights into the cardiovascular system. It continuously provides physiological parameters that can be utilized to estimate health information, including heart rate, respiratory rate, and oxygen saturation, contributing to the comprehensive nature of these health-focused wearables. PPG signal plays a central role in these estimations due to its non-invasive nature and cost-effective implementation on devices. PPG works by emitting a light signal onto the user's skin surface and capturing its reflection/transmission, which varies proportionally to the blood volume flow in the tissue.

A major challenge impacting PPG's on-device and real-world performance is its susceptibility to noise, including motion artifacts (Chatterjee et al., 2022), which can distort the signal's morphological properties and result in incorrect estimation of the aforementioned physiological variables. Given the potentially life-threatening consequences associated with inaccurate assessments derived from these signals, such unreliable performance would be highly inadequate to real-world applications. The presence of noisy signal sections is the main driving force behind the development of Signal Quality Assessment (SQA) techniques. This is vital to prevent misinterpretation by distinguishing between clinically validated trustworthy and untrustworthy segments in PPG. In essence, to enhance the reliability of such applications, a Signal Quality Classifier (SQC) step is commonly employed, enabling the differentiation of high-quality signal segments, i.e., suitable for physiological variables' estimation, from low-quality ones.

## 1.1 RELATED WORK

Given the significance of classifying signal segments into reliable or unreliable, several researchers have dedicated valuable efforts to developing classification techniques for signal quality assessment, as reviewed by Gambarotta et al. (2016). For example, Elgendi (2016) proposed a technique employing indices to assess quality. Selvaraj et al. (2011) presented a statistical method involving the calculation of kurtosis and Shannon Entropy to identify motion artifacts and noise in PPG data. Li & Clifford (2012) introduced an alternate statistical approach, leveraging dynamic time warping to stretch each heartbeat for alignment with a dynamic template. This method incorporates various features associated with signal quality. Regarding frequency domain, (**?**) proposes a classification method making use of Deep Fourier Magnitude Spectrum. In the classification phase of the approach described in (Li & Clifford, 2012), a multi-layer perceptron is utilized to comprehend the correlations among parameters in the context of both high- and low-quality pulses.

Sun et al. (2012) introduced an approach that exploits the morphological features of the signal for evaluating its quality. Li et al. (2011) identified four waveform characteristics to evaluate signal quality through the application of a decision tree. Similarly, Sukor et al. (2011) utilized a basic decision-tree classifier to determine, with pulse-by-pulse precision, whether a particular pulse is suitable for use or not. Naeini et al. (2019) introduced one of the earliest ML-based methods to categorize the signal into 'reliable' or 'unreliable' categories. Recently, Freitas et al. (2023a) and Freitas et al. (2023b) introduced a SQA technique that transforms PPG signals into two-dimensional representations and subsequently employs a vision transformer to evaluate their quality. (**?**) proposes an ensemble of local magnitude comparison-based feature descriptors for subsequently feeding into a linear classifier.

Additionally, the work in (Silva et al., 2024) explored the usage of attention-based approaches of (Bahdanau et al., 2015; Luong et al., 2015; Vaswani et al., 2017) in conjunction with a Convolutional-based strategy to find an efficient Signal Quality Classifier (SQC) under strict trade-offs among memory and classification performance, as well as considerations regarding overall power consumption. In the present work, we enhance the performance and efficiency of CNN-based approach to PPG SQA by jointly leveraging differential and integral attention layers (Ye et al., 2024) prior to the final dense layer, as a way to develop an effective SQC capable of being deployed in real-time HMA of resource-constrained embedded devices.

## 2 PROPOSED METHOD

### 2.1 OVERVIEW

In the present work, we concern ourselves with the binary quality classification of online unidimensional PPG signal segments, which is the predominant case for wearable-based PPG sensing. Our aim is to enhance the capabilities of a compact CNN using a differential attention mechanism (Ye et al., 2024) for quality classification of PPG sensor data, a functionality widely used in wearable apps for heart rate and sleep assessment.

Initially, CNNs were proposed for computer vision tasks, automatically extracting local and global features from images or frames for classification. However, they have expanded to other domains, including time series classification, such as biomedical ones (Lucafo et al., 2022). Meanwhile, attention mechanisms originated in natural language processing to overcome the issue of vanishing gra-

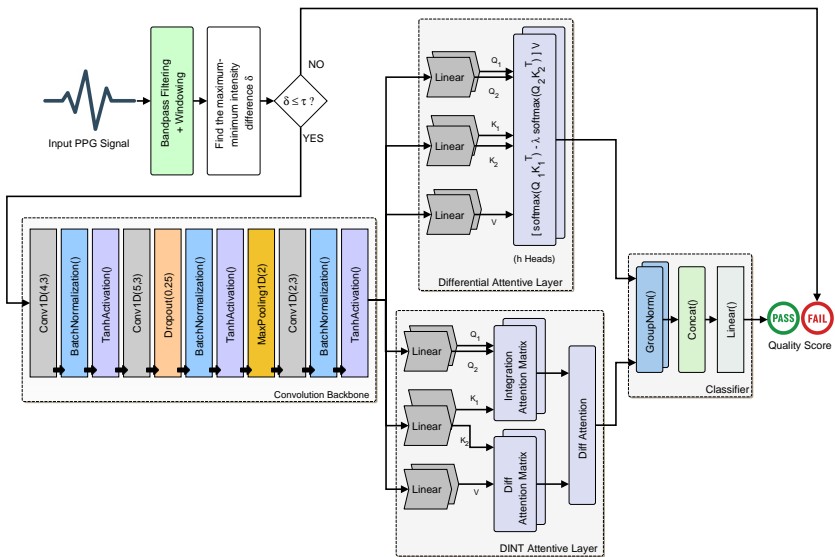

Figure 1: Overview of the proposed PPG classification method.

dients in long-term dependency learning of sequential models (Bahdanau et al., 2015), like RNNs. They have achieved state-of-the-art performance across various domains, including LLMs (Brown et al., 2020) and vision-related tasks (Freitas et al., 2023b;a).

We utilize Neural Architecture Search (NAS) techniques and attention mechanisms to identify neural network solutions that are lightweight yet deliver competitive performance within strict memory and computational constraints, as shown in Figure 1. Our baseline CNN, also detailed in Figure 1, was discovered through a NAS process similar to (Lima et al., 2023), specifically tailored for PPG signal quality assessment rather than systolic peak classification. The additional attention mechanisms we exploit — Differential Attention and Differential-Integral Attention Layers — will be discussed in Sections 2.4 and 2.5.

## 2.2 PREPROCESSING

Regarding the preprocessing used in this work, we start from the Discrete 1-D PPG vector

$$\mathbf{Y} = \{y_k\}_0^{N_{samples}-1} \in \mathbb{R}^{N_{samples}}, \tag{1}$$

collected with sampling frequency $f_s = 25$Hz.

The considered vector is subject to a windowing operation, resulting in segments of 9 seconds with 3 seconds overlap on each edge, after which we apply an order-2 zero-phase Butterworth bandpass filter with cutoff frequencies of 0.8 and 4.5Hz. The lower cutoff frequency aims to remove the baseline wandering while the higher frequency removes environmental noise of high frequency nature.

The segments are, then, further converted to a dataset $\mathcal{Y} \in \mathbb{R}^{N \times L}$ composed of $N$ non-overlapping segments of 3-seconds with signal length of $L$ samples, in the form

$$\mathcal{Y} = \{Y_k\}_{k=0}^{N-1} \text{ ,with } Y_k = \{y_{N \cdot k+j}\}_{j=0}^{L-1} \tag{2}$$

with $L = 3 \cdot f_s = 75$ samples, and $N = \left\lfloor \dfrac{N_{samples}}{L} \right\rfloor$, for $N_{samples}$ the total number of samples.

Each segment is associated with a corresponding binary label, with 0 meaning low-quality (unreliable) PPG data and 1 meaning high-quality (reliable) PPG data. A cardiologist provided a sample-by-sample ground truth of 0s and 1s for the entire dataset. The label of the entire 3-seconds segment (= 75 samples) could be computed by evaluating the prevalence of samples labelled as 1. If it is equal or higher than 50%, then the entire segment is considered as having a label 1. Otherwise, the segment label is assigned to 0.

## 2.3 Two-Stage Signal Classification

A proposed Two-Stage Signal Quality Classification (SQC) pipeline, inspired by (Lucafo et al., 2022), was considered as a starting point. It has a hierarchical structure composed of an initial decision stage with the purpose of discarding PPG windows that are most likely to be of low quality based on the empirical observation of a given valid signal interval. The second stage of the SQC consists of the classification of the PPG window by a designed Attentive-CNN.

The idea behind the discriminator in the first stage is to analyze each PPG window provided as input and decide, based on its difference between maximum and minimum values $\delta$, if the whole window will be considered to be of low quality or possibly a high quality one, as previously presented in the Materials and Methods section. This decision is assisted by the threshold $\tau$ chosen from the earlier mentioned methods. The results regarding the most empirically reliable thresholds will be further discussed in the Results section.

The principle of the Attentive-CNN of the second stage is to process the data considered to be of possibly high quality based on the discrimination stage described above. The stage acts on a more limited interval, when a more refine analysis is requested to detect subtle waveform differences of the signal, in order to classify it. Such analysis was found to be more suitable for learning-based methods.

In this work, we also investigate the impact of this two-stage classification on three state-of-the-art attention-based PPG classification methods: Additive Attention Layer (AAL), Dot Product Attention Layer (DPAL) and Non-Scaled Dot Product Attention Layer (NSDPAL), described on (Silva et al., 2024), besides the composite classifiers Differential Attention and Differential-Integral Attention Layers to be defined in Sections 2.4 and 2.5, respectively.

## 2.4 Differential Attention

The Differential (DIFF) attention mechanism (Ye et al., 2024) works through mapping query, key, and value vectors to the final outputs, using query and key vectors to compute intermediate attention scores, which are then used to compute a final weighted sum of value vectors. The most crucial step of the proposed design is using paired softmax functions for noise canceling in the intermediate attention scores. The last tanh layer of the CNN from first stage produces vectors $X \in R^{N \times d_{model}}$, with $d_{model}$ the hidden dimension of the model, which are considered inputs to the differential attention mechanism. Such inputs are initially projected to corresponding query vectors $Q_1, Q_2 \in R^{N \times d}$, key vectors $K_1, K_2 \in R^{N \times d}$, and value vectors $V \in R^{N \times 2d}$. The dimension factor $d$ is given by $d = d_{model}/2h$, with $h$ the number of attention heads, here set to $h = 1$. Afterwards, the differential attention operator DiffAttn$(\cdot)$ calculates the respective outputs as:

$$[Q_1; Q_2] = XW^Q, \ [K_1; K_2] = XW^K, V = XW^V \tag{3}$$

$$\text{DiffAttn}(X) = \left( \sigma \left( \frac{Q_1 K_1^T}{\sqrt{d}} \right) - \lambda \sigma \left( \frac{Q_2 K_2^T}{\sqrt{d}} \right) \right) V, \tag{4}$$

where $\sigma(\cdot)$ is the softmax function

$$\sigma(z_i) = \frac{e^{z_i}}{\sum_{j=1}^{K} e^{z_j}}, \quad \text{for } i = 1, 2, \ldots, K \tag{5}$$

with $W^Q, W^K, W^V \in R^{d_{model} \times 2d}$ being trainable parameters, while $\lambda$ is a learnable scalar value. With the purpose of synchronizing the learning dynamics, the scalar $\lambda$ is reparameterized as:

$$\lambda = e^{(\lambda_{q_1} \cdot \lambda_{k_1})} - e^{(\lambda_{q_2} \cdot \lambda_{k_2})} + \lambda_{init} \tag{6}$$

where $\{\lambda_{q_1}, \lambda_{k_1}, \lambda_{q_2}, \lambda_{k_2}\} \subset \mathbb{R}^d$ are learnable vectors, and $\lambda_{init} \in [0, 1]$ is a constant used for the initialization of $\lambda$. The best value of $\lambda_{init}$ was empirically searched along the range of $[0, 1]$, reporting each tested value in the metric tables. The experimental results show that the performance varies slightly along different values of initialization.

Thus, the Differential attention mechanism computes the difference between two softmax attention functions, as a way of eliminating attention noise. The proposed idea can be shown to be analogous to that of differential amplifiers (Laplante, 2018), more commonly studied in the electrical engineering domain, in which the difference between two signals being taken as output, as a way to do away with the so-called common-mode noise of the input.

## 2.5 DIFFERENTIAL INTEGRAL ATTENTION

Differential Integral (DINT) attention (Cang et al., 2025) is an extension of the differential (DIFF) attention which introduces an additional integral mechanism, that enhances the model's capability of extracting globally relevant information while retaining numerical stability through the normalization of rows in the corresponding final attention matrix. The attention matrix $A_1$ of the signal is calculated using $Q_1$ and $K_1$:

$$A_1 = \sigma \left( \frac{Q_1 K_1^T}{\sqrt{d}} \right) \tag{7}$$

The integral mechanism computes the global attention importance scores through averaging the signal attention weights across each column:

$$G = \frac{1}{N} \sum_{m=1}^{N} A_1 \left[ m, n \right], \tag{8}$$

with $G \in \mathbf{R}^{1 \times N}$ then being expanded for matching the dimensions of the differential mechanism:

$$G_{expanded} = \text{repeat}(G, N), \tag{9}$$

with $G_{expanded} \in \mathbf{R}^{N \times N}$ being constructed by repeating G along N rows. Thus, the full Differential-Integral attention mechanism, represented by the DINT operator, calculates the output as:

$$DINT(X) = (A_{diff} + \gamma G_{expanded}) V, \tag{10}$$

with $\gamma$ being a scalar, $A_{diff}$ being the differential attention mechanism, and $G_{expanded}$ being the expanded matrix of global importance scores. Usually, $\lambda$ and $\gamma$ are set to equal values for ensuring that the resulting attention matrix $A_{final}$ has rows summing up to 1. This normalization of rows assures numerical stability and consistency along the model, what increases data stability across layers. This unified configuration follows the method of parameterization used in the original DIFF Transformer, as a means of further increasing stability during training.

In the present work, we proposed to enhance one-stage and two-stage CNN-based PPG classification (Lucafo et al., 2022) by adopting both Differential and Differential-Integral attention mechanisms replacing the GlobalAveragePooling1D layer of the NAS-optimized CNN. We hereby refer to the two composite classifiers as Differential Dot-Product Attention Layer (DFPAL) and Differential-Integral Dot-Product Attention Layer (DINTAL), respectively.

## 3 EXPERIMENTAL SETUP

To assess the performance of the proposed architectures outlined in Figure 3, we compare the identified neural network classifiers against the state-of-the-art SQC method using the same evaluation process on four distinct datasets:

- GW5: Comprising 149 sessions lasting approximately 35 minutes each, with 119 utilized for training and 30 for testing. PPG samples were obtained with Samsung Galaxy Watch 5.
- GW6: Consisting of 94 sessions lasting around 35 minutes each, with 75 dedicated to training and 19 assigned to testing. PPG readings were collected utilizing a Samsung Galaxy Watch 6.
- GW7: Including 37 sessions lasting approximately 35 minutes each, with 30 allocated for training and 7 designated for testing. PPG readings were acquired using a Samsung Galaxy Watch 7.
- RING: Comprised of 73 sessions lasting roughly 35 minutes each, with 59 reserved for training and 14 set aside for testing. PPG readings were obtained using a Samsung Galaxy Ring.

For each Galaxy Watch subject, the PPG sensor was carefully positioned on the wrist over the radial artery to optimize signal acquisition. For Galaxy Ring subjects, the device was inserted in the finger which provided the best adherence to the skin. Data was gathered under resting conditions, with participants instructed to sit or lie down in a quiet environment. The quality of the signals

in the datasets was manually labeled by a cardiologist expert, taking into account the waveform characteristics and the dataset's purpose of measuring Interbeat Intervals (IBIs).

The PPG signals were partitioned into non-overlapping 3-second windows, each containing 75 samples. To prepare for the learning phase, each segment was labeled depending on whether the proportion of high-quality samples exceeded a given threshold: segments with more than or equal to 50% high-quality signal were classified as 'reliable', while others were deemed 'unreliable'.

For the experiments, we separated the data into training and test sets. Validation subsets were randomly selected from the training set. During training, we utilized data and corresponding labels from 64% of the subjects in each assessment group. For validation, we chose data from 16% of the subjects in each group. The remaining 20% of subjects in each group were reserved for testing the proposed methods and the entire pipeline. It is crucial to note that the signal segments used in the experiments were organized according to the subject they belonged to. This ensured that there was no overlap between training and testing sets, preventing potential training biases from influencing testing results. For our proposed method, we concurrently performed an ablation of the $\lambda_{init}$ parameters across the range of values in $\{0.1, 0.2, 0.3, 0.4, 0.5, 0.6, 0.7, 0.8, 0.9, 0.99\}$.

For implementing the DL models, we used Keras 2.9.0 and Python 3.8.17. For the remaining classifiers, we used scikit-learn 1.3.0. All models underwent 5 trials of 100 training epochs each, with the resulting test set performance metrics being averaged across all trials. To gauge the performance of both the proposed and state-of-the-art methods, we compared the predicted quality indices with the pre-labeled indices provided in the benchmark dataset using Accuracy, Precision, Recall, F1-Score, AUC, Coverage, Matthews Correlation Coefficient (MCC), and Cohen's Kappa metrics. For more information on these metrics, refer to (Dalianis & Dalianis, 2018).

## 4 EXPERIMENTAL RESULTS

We have implemented 10 models adopting additive attention (AAL), dot-product attention (DPAL), non-scaled dot-product attention (NSDPAL), differential dot-product attention (DFPAL) and differential-integral dot-product attention (DINTAL) mechanisms in Attentive-CNN or Two-Stage architectures. Then, their performance was compared against 6 benchmarks based on convolutional neural networks (Lucafo1 and Lucafo 2), descriptor features (CASLBP SGD, CASLTP SGD and LBP SGD) and sets of rules (Hao & Bo), considering the cardiologist label as the ground-truth.

The average performance metrics for our proposed method with DFPAL, as well as baseline out-of-the-box classifiers and state-of-the-art attention-based PPG classification methods, are shown in Table 1. It can be observed that architectures adopting DINTAL and DFPAL mechanisms generally outperform other methods in terms of accuracy, F1-score, AUC, MCC and Cohen's Kappa.

Regarding model size, Figure 2 shows that the required increase in the model total size of the proposed composite DFPAL and DINTAL methods is not proportionally substantial when compared to the attention-based state-of-the-art comparison methods. The increase in model size is significative when compared with lower-performing methods, such as LBP, CASLBP and CASLTP. When memory and computational burden restrictions are too strict, such memory-performance trade-offs may have to be taken into consideration.

Analysing the impact of $\lambda_{init}$ parameter on the test set metrics of the model, Figures 3, 4, 5, and 6 show how the performance metrics vary with such free parameter in each dataset. The values of accuracy, precision, recall and F1-score are generally stable, while AUC, MCC and Cohen's Kappa may vary substantially. Two-Stage architectures adopting DFPAL and DINTAL mechanisms are considerably more sensitive to $\lambda_{init}$ selection than Attention-CNN architectures. Hence, fine-tuning step can be critical for the desirable high performance in such cases.

## 5 CONCLUSIONS

Wearable health monitoring apps benefit greatly from PPG technology, especially when worn on the wrist. Its configuration is simple, convenient, low-complexity, and cost-effective. Despite these advantages, PPG signals are vulnerable to significant degradation due to various factors, primarily motion artifacts, which can distort the waveform morphology, impacting subsequent signal analysis.

| Dataset | Model | Accuracy | Precision | Recall | F1-Score | AUC | Coverage | MCC | Kappa |
|---|---|---|---|---|---|---|---|---|---|
| GW5 | Att-CNN+DFPAL | 0.9736 | 0.9797 | 0.9964 | 0.9864 | 0.7330 | 0.9830 | 0.5714 | 0.5562 |
| | Att-CNN+DINTAL | **0.9737** | 0.9796 | 0.9961 | **0.9865** | 0.7318 | 0.9825 | 0.5740 | 0.5584 |
| | Two-Stage+DFPAL | 0.9709 | **0.9887** | 0.9914 | 0.9849 | **0.8406** | 0.9761 | 0.5829 | 0.5778 |
| | Two-Stage+DINTAL | 0.9695 | 0.9873 | 0.9871 | 0.9842 | 0.8265 | 0.9697 | **0.6020** | **0.5979** |
| | Att-CNN+AAL | 0.9722 | 0.9765 | 0.9951 | 0.9857 | 0.6904 | 0.9808 | 0.5285 | 0.4937 |
| | Att-CNN+DPAL | 0.9732 | 0.9783 | 0.9942 | 0.9862 | 0.7148 | 0.9781 | 0.5582 | 0.5352 |
| | Att-CNN+NSDPAL | 0.9734 | 0.9791 | 0.9936 | 0.9863 | 0.7249 | 0.9767 | 0.5676 | 0.5500 |
| | Two-Stage+AAL | 0.9707 | 0.9803 | 0.9895 | 0.9848 | 0.7396 | 0.9715 | 0.5462 | 0.5325 |
| | Two-Stage+DPAL | 0.9632 | 0.9816 | 0.9803 | 0.9808 | 0.7523 | 0.9614 | 0.5124 | 0.4911 |
| | Two-Stage+NSDPAL | 0.9708 | 0.9801 | 0.9897 | 0.9849 | 0.7376 | 0.9719 | 0.5466 | 0.5333 |
| | Lucafo1 (Lucafo et al., 2022) | 0.9720 | 0.9794 | 0.9918 | 0.9855 | 0.7284 | 0.9746 | 0.5526 | 0.5398 |
| | Lucafo2 (Lucafo et al., 2022) | 0.9677 | 0.9778 | 0.9890 | 0.9833 | 0.7056 | 0.9736 | 0.4866 | 0.4622 |
| | CASLBP SGD (Garcia Freitas et al., 2025) | 0.9647 | 0.9710 | 0.9930 | 0.9819 | 0.6174 | 0.9841 | 0.3584 | 0.3238 |
| | CASLTP SGD (Garcia Freitas et al., 2025) | 0.9631 | 0.9643 | **0.9986** | 0.9812 | 0.5260 | **0.9967** | 0.0774 | 0.0711 |
| | LBP SGD (Garcia Freitas et al., 2025) | 0.9638 | 0.9700 | 0.9932 | 0.9814 | 0.6033 | 0.9854 | 0.2944 | 0.2740 |
| | Hao & Bo (Hao & Bo, 2021) | 0.5808 | 0.9702 | 0.5823 | 0.7278 | 0.5619 | 0.5776 | 0.0477 | 0.0209 |
| GW6 | Att-CNN+DFPAL | 0.7212 | 0.7351 | 0.9985 | 0.8147 | 0.6378 | 0.9806 | 0.3342 | 0.2841 |
| | Att-CNN+DINTAL | 0.7194 | 0.7571 | 0.9982 | 0.8104 | 0.6484 | 0.9825 | 0.3367 | 0.2922 |
| | Two-Stage+DFPAL | **0.8967** | 0.9301 | 0.9234 | 0.9192 | **0.8925** | 0.6468 | **0.7786** | **0.7776** |
| | Two-Stage+DINTAL | 0.8952 | 0.9274 | 0.9227 | 0.9183 | 0.8897 | 0.6475 | 0.7733 | 0.7733 |
| | Att-CNN+AAL | 0.6915 | 0.7033 | 0.9511 | 0.7996 | 0.5909 | 0.8857 | 0.2703 | 0.1910 |
| | Att-CNN+DPAL | 0.6598 | 0.6563 | 0.9832 | 0.7871 | 0.5345 | 0.9583 | 0.1713 | 0.0853 |
| | Att-CNN+NSDPAL | 0.6891 | 0.6911 | 0.9691 | 0.8017 | 0.5806 | 0.9110 | 0.2528 | 0.1737 |
| | Two-Stage+AAL | 0.8967 | 0.9183 | 0.9210 | **0.9194** | 0.8872 | 0.6419 | 0.7765 | 0.7755 |
| | Two-Stage+DPAL | 0.8931 | 0.9158 | 0.9174 | 0.9165 | 0.8837 | 0.6409 | 0.7684 | 0.7679 |
| | Two-Stage+NSDPAL | 0.8962 | 0.9172 | 0.9210 | 0.9190 | 0.8866 | 0.6423 | 0.7748 | 0.7745 |
| | Lucafo1 (Lucafo et al., 2022) | 0.7006 | 0.7069 | 0.9427 | 0.8035 | 0.6068 | 0.8657 | 0.2847 | 0.2318 |
| | Lucafo2 (Lucafo et al., 2022) | 0.8960 | 0.9266 | 0.9099 | 0.9179 | 0.8906 | 0.6284 | 0.7767 | 0.7758 |
| | CASLBP SGD (Garcia Freitas et al., 2025) | 0.8487 | 0.8709 | 0.8965 | 0.8835 | 0.8302 | 0.6586 | 0.6688 | 0.6681 |
| | CASLTP SGD (Garcia Freitas et al., 2025) | 0.6396 | 0.6396 | **1.0000** | 0.7802 | 0.5000 | **1.0000** | 0.0000 | 0.0000 |
| | LBP SGD (Garcia Freitas et al., 2025) | 0.8545 | 0.8794 | 0.8954 | 0.8873 | 0.8386 | 0.6514 | 0.6824 | 0.6821 |
| | Hao & Bo (Hao & Bo, 2021) | 0.4109 | **0.9395** | 0.0844 | 0.1549 | 0.5374 | 0.0575 | 0.1542 | 0.0552 |
| GW7 | Att-CNN+DFPAL | 0.9555 | 0.9551 | **1.0000** | 0.9762 | 0.7473 | **0.9942** | 0.6738 | 0.6325 |
| | Att-CNN+DINTAL | 0.9533 | 0.9547 | 0.9996 | 0.9750 | 0.7444 | 0.9709 | 0.6569 | 0.6202 |
| | Two-Stage+DFPAL | 0.9623 | **0.9765** | 0.9895 | 0.9796 | **0.8625** | 0.9333 | 0.7405 | 0.7356 |
| | Two-Stage+DINTAL | **0.9626** | 0.9752 | 0.9887 | **0.9797** | 0.8551 | 0.9327 | **0.7465** | **0.7416** |
| | Att-CNN+AAL | 0.9532 | 0.9526 | 0.9985 | 0.9750 | 0.7325 | 0.9589 | 0.6527 | 0.6044 |
| | Att-CNN+DPAL | 0.9521 | 0.9520 | 0.9980 | 0.9745 | 0.7290 | 0.9590 | 0.6444 | 0.5982 |
| | Att-CNN+NSDPAL | 0.9503 | 0.9491 | 0.9993 | 0.9735 | 0.7117 | 0.9633 | 0.6276 | 0.5694 |
| | Two-Stage+AAL | 0.9599 | 0.9759 | 0.9805 | 0.9782 | 0.8600 | 0.9191 | 0.7380 | 0.7367 |
| | Two-Stage+DPAL | 0.9608 | 0.9677 | 0.9902 | 0.9788 | 0.8176 | 0.9361 | 0.7255 | 0.7153 |
| | Two-Stage+NSDPAL | 0.9607 | 0.9754 | 0.9819 | 0.9786 | 0.8580 | 0.9209 | 0.7416 | 0.7393 |
| | Lucafo1 (Lucafo et al., 2022) | 0.9536 | 0.9616 | 0.9888 | 0.9750 | 0.7822 | 0.9407 | 0.6750 | 0.6574 |
| | Lucafo2 (Lucafo et al., 2022) | 0.9607 | 0.9698 | 0.9879 | 0.9788 | 0.8284 | 0.9320 | 0.7290 | 0.7201 |
| | CASLBP SGD (Garcia Freitas et al., 2025) | 0.9295 | 0.9297 | 0.9984 | 0.9628 | 0.5938 | 0.9824 | 0.3971 | 0.2901 |
| | CASLTP SGD (Garcia Freitas et al., 2025) | 0.9231 | 0.9231 | 0.9993 | 0.9596 | 0.5517 | 0.9905 | 0.1896 | 0.1535 |
| | LBP SGD (Garcia Freitas et al., 2025) | 0.9296 | 0.9292 | 0.9992 | 0.9629 | 0.5904 | 0.9838 | 0.3580 | 0.2760 |
| | Hao & Bo (Hao & Bo, 2021) | 0.6383 | 0.9302 | 0.6536 | 0.7678 | 0.5634 | 0.6428 | 0.0739 | 0.0518 |
| RING | Att-CNN+DFPAL | 0.8794 | 0.8761 | **0.9940** | 0.9279 | 0.7381 | 0.9017 | 0.6105 | 0.5707 |
| | Att-CNN+DINTAL | 0.8784 | 0.8760 | 0.9934 | 0.9275 | 0.7372 | 0.9013 | 0.6082 | 0.5666 |
| | Two-Stage+DFPAL | 0.9030 | 0.9479 | 0.9349 | 0.9380 | 0.8663 | 0.7803 | 0.7154 | 0.7152 |
| | Two-Stage+DINTAL | **0.9045** | **0.9494** | 0.9374 | **0.9390** | **0.8679** | 0.7824 | **0.7183** | **0.7181** |
| | Att-CNN+AAL | 0.8776 | 0.8706 | 0.9918 | 0.9272 | 0.7260 | 0.8947 | 0.6043 | 0.5526 |
| | Att-CNN+DPAL | 0.8764 | 0.8696 | 0.9912 | 0.9264 | 0.7240 | 0.8950 | 0.6004 | 0.5500 |
| | Att-CNN+NSDPAL | 0.8794 | 0.8723 | 0.9915 | 0.9281 | 0.7306 | 0.8924 | 0.6115 | 0.5632 |
| | Two-Stage+AAL | 0.8936 | 0.9492 | 0.9134 | 0.9308 | 0.8673 | 0.7555 | 0.7033 | 0.7003 |
| | Two-Stage+DPAL | 0.8958 | 0.9412 | 0.9252 | 0.9331 | 0.8568 | 0.7718 | 0.6990 | 0.6980 |
| | Two-Stage+NSDPAL | 0.8917 | 0.9471 | 0.9133 | 0.9297 | 0.8631 | 0.7572 | 0.6973 | 0.6941 |
| | Lucafo1 (Lucafo et al., 2022) | 0.8836 | 0.8787 | 0.9883 | 0.9302 | 0.7446 | 0.8831 | 0.6262 | 0.5860 |
| | Lucafo2 (Lucafo et al., 2022) | 0.8994 | 0.9417 | 0.9297 | 0.9355 | 0.8592 | 0.7753 | 0.7081 | 0.7066 |
| | CASLBP SGD (Garcia Freitas et al., 2025) | 0.8424 | 0.8602 | 0.9545 | 0.9048 | 0.6936 | 0.8713 | 0.4753 | 0.4528 |
| | CASLTP SGD (Garcia Freitas et al., 2025) | 0.8148 | 0.8218 | 0.9821 | 0.8935 | 0.5930 | **0.9421** | 0.2171 | 0.2110 |
| | LBP SGD (Garcia Freitas et al., 2025) | 0.8492 | 0.8704 | 0.9493 | 0.9081 | 0.7164 | 0.8563 | 0.5068 | 0.4919 |
| | Hao & Bo (Hao & Bo, 2021) | 0.6180 | 0.8368 | 0.6379 | 0.7239 | 0.5917 | 0.5985 | 0.1536 | 0.1394 |

Table 1: Performance comparison of the best DFPAL and DINTAL models, with and without the Two-Stage Classification pipeline, alongside baseline attention layers (AAL, DPAL, NSDPAL) and other state-of-the-art methods on benchmark datasets. All methods were trained for five trials over 1000 epochs, with test metrics averaged across trials.

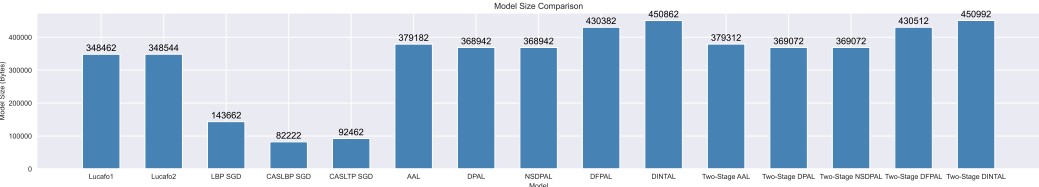

Figure 2: Model Size Comparison across proposed attention-based solutions and state-of-the-art comparison methods. Both one-stage and two-stage compositions are consdered.

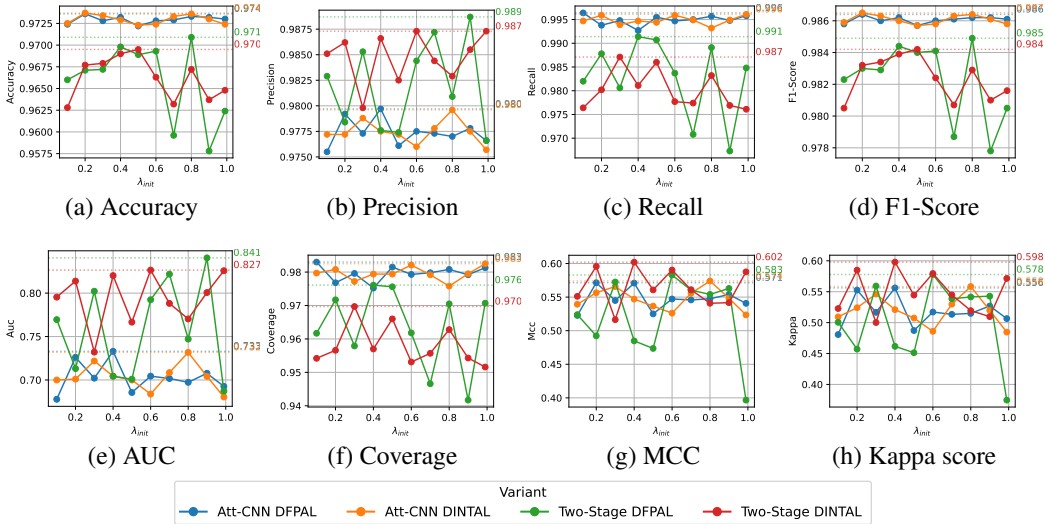

Figure 3: Performance comparison of DFPAL and DINTAL, with and without Two-Stage Classification pipeline, across several values of $\lambda_{init}$ on **GW5** PPG dataset. All the methods were trained for 5 trials and 1000 epochs. The test set performance metrics were then averaged across all trials.

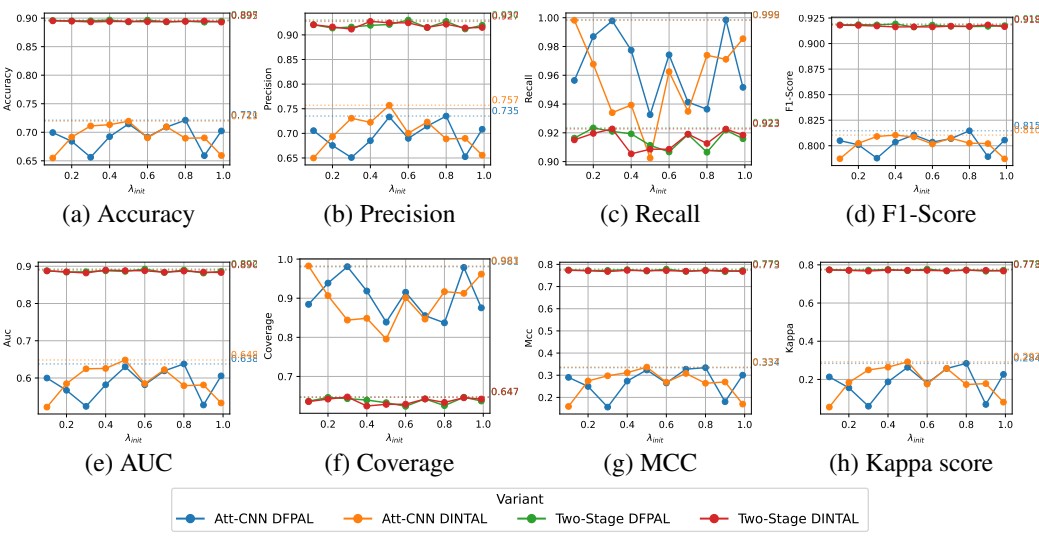

Figure 4: Performance comparison of DFPAL and DINTAL, with and without Two-Stage Classification pipeline, across several values of $\lambda_{init}$ on **GW6** PPG dataset. All the methods were trained for 5 trials and 1000 epochs. The test set performance metrics were then averaged across all trials.

Erroneous decisions and misjudgments resulting from unreliable signals are unacceptable for HMA. Thus, assessing PPG signal quality is essential to prevent misinterpretation and differentiate between reliable and noisy signals. To address this, in this study we leveraged ML models to develop robust and high-performing models for classifying PPG segments with the purpose of estimating SQA.

Efficient Differential and Differential-Integral Attentive and Convolutional (CNN)-based approaches have been explored in this work to classify PPG signal segments as usable or not, further verifying their effectiveness by applying them to four distinct expert-annotated photoplethysmographic datasets. The task they performed is framed as a binary classification problem and plays a vital role in enabling reliable HMA, particularly in wearable devices. By employing NAS-based techniques, we discovered a baseline CNN architecture that is small enough to be deployed on embedded devices, facilitating real-time wearable HMA. Furthermore, we demonstrate that incorpo-

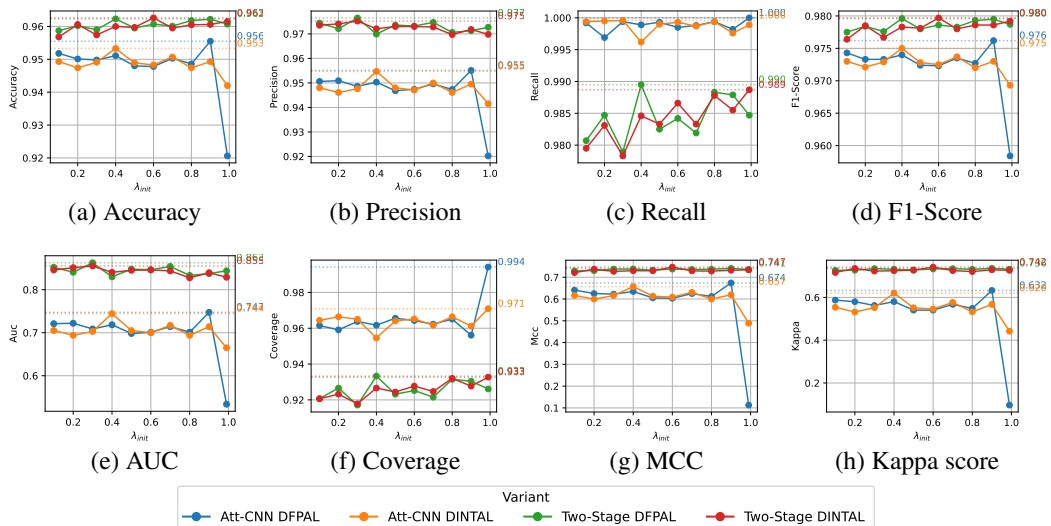

Figure 5: Performance comparison of DFPAL and DINTAL, with and without Two-Stage Classification pipeline, across several values of $\lambda_{init}$ on **GW7** PPG dataset. All the methods were trained for 5 trials and 1000 epochs. The test set performance metrics were then averaged across all trials.

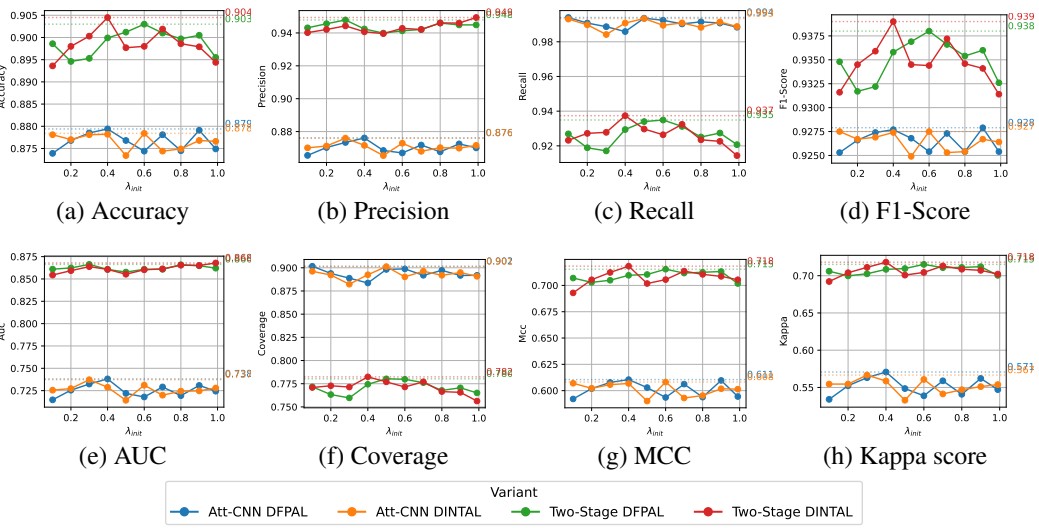

Figure 6: Performance comparison of DFPAL and DINTAL, with and without Two-Stage Classification pipeline, across several values of $\lambda_{init}$ on **RING** PPG dataset. All the methods were trained for 5 trials and 1000 epochs. The test set performance metrics were then averaged across all trials.

rating an additional fine-tuned differential attention layer enhances the performance of the baseline CNN, effectively boosting classification metrics without incurring significant additional computational costs.

## DECLARATION OF GENERATIVE AI AND AI-ASSISTED TECHNOLOGIES IN THE WRITING

During the preparation of this work the authors used ChatGPT™ and Grammarly™ to review English usage and grammatical correction. After using these tools, the authors reviewed and edited the content as needed and take full responsibility for the content of the publication.

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
