# OpenReview forum: "Differentially and Integrally Attentive Convolutional-based Photoplethysmography Signal Quality Classification"
_ICLR.cc/2026/Conference — ICLR 2026 Conference Withdrawn Submission_

### Official Review · Reviewer_TDdY · 2025-10-28

**Soundness:** 2
**Presentation:** 2
**Contribution:** 2
**Rating:** 2
**Confidence:** 4

**Summary:**

The paper proposes a two-stage framework to enhance CNN-based PPG signal quality assessment using two attention mechanisms: differential attention and differential integral attention. These mechanisms aim to improve the model’s ability to focus on informative waveform features by minimizing discrepancies in the attention mechanism. The methods are evaluated on four devices/datasets, showing improved performance over selected baselines.

**Strengths:**

- Signal quality of PPG measurements is an important problem that can affect performance of downstream tasks. This paper tackles this challenge by evaluating many different models.
- The datasets used in the paper consists of different devices and the labels have been annotated by a cardiologist. As a result, both the dataset quality and the insights derived from the analysis have the potential to be highly meaningful and impactful.

**Weaknesses:**

**Explanation of core contributions and related work**: The paper does not adequately explain the core contributions. In particular:
- **(1) Difference from previous work:** While the related work describes prior work in PPG SQA, it does not clarify how the proposed work is different
- **(2) Technical contribution:** The technical contribution of the paper remains unclear because core idea of this work and text focus on differential and differentially-integral attention. Both these approaches are proposed by other works [1, 2]. Instead the paper needs to clarify the core contribution, perhaps it lies in a novel integration, adaptation, or application of these methods.
- **(3) Significance to ICLR:**  I believe that paper's contribution to ICLR is fairly limited because of the following reasons: (a) The technical contributions are limited as explained above. (b) On the other hand, the application of the paper is limited to SQA using PPG. While application-oriented papers can fit well within ICLR, previous PPG-related studies [3, 4] have demonstrated stronger methodological and conceptual contributions.

**Experimental Results**: The difference between using DPFAL and DINT is negligible (e.g., 0.8967 and 0.8952). Therefore, I wonder if this difference is meaningful in any way. Adding confidence intervals or some statistical measure will provide better information.

Figure 1. indicates DINT Attentive layer and Differential Attentive Layer are used together. However, from the writing in Section 2.5, DFPAL refers to Differential Attentive Layer and DINTAL refers to DINT Attentive Layer. Is this a discrepancy?

It is challenging to interpret which model performs best for the task because of two reasons. (a) While I appreciate the implementation of several models to evaluate signal quality, the paper should include a more focused analysis and provide guidance on which models to choose. At the moment, it seems that the number of proposed models is more than the number of baselines. (b) Many of these metrics can be moved to the appendix and the main paper can only describe metrics that are relevant to the problem (e.g., AUC and MCC).

**Data Provenance**: Several important details about the dataset are missing, including participant demographics, the total number of signal segments, and the ratio of reliable to unreliable signals.

**Results, Discussion, and Writing**: The results section provides only a superficial overview without emphasizing the most important findings. Moreover, there is no discussion that draws meaningful insights or interprets the implications of the results. The writing in these sections needs significant improvement to better communicate the key takeaways and their relevance. Additionally, the paper should include a dedicated Limitations section to acknowledge potential shortcomings and outline directions for future work.

**Efficiency Analysis**: One of the stated motivations of this work is its potential for deployment on resource-constrained devices. However, the evidence supporting this claim is limited. The paper only reports model sizes, which provides an incomplete picture of efficiency. Additional metrics such as latency and throughput are needed to thoroughly assess the model’s suitability for resource-limited environments.

[1] Ye, T., Dong, L., Xia, Y., Sun, Y., Zhu, Y., Huang, G., & Wei, F. (2024). Differential transformer. arXiv preprint arXiv:2410.05258.

[2] Cang, Y., Liu, Y., Zhang, X., Zhao, E., & Shi, L. (2025). Dint transformer. _arXiv preprint arXiv:2501.17486_.

[3] Abbaspourazad, S., Elachqar, O., Miller, A. C., Emrani, S., Nallasamy, U., & Shapiro, I. (2023). Large-scale training of foundation models for wearable biosignals. _arXiv preprint arXiv:2312.05409_.

[4] Pillai, A., Spathis, D., Kawsar, F., & Malekzadeh, M. (2024). Papagei: Open foundation models for optical physiological signals. _arXiv preprint arXiv:2410.20542_.

**Questions:**

- **Data Availability**: Indicate if the dataset will be released with paper upon publication.
- Please fix missing citations throughout the paper (related work question marks and missing citations for baselines).
- Is $\delta$ simply the subtraction the highest and lowest magnitude?
- **Method**: Figure 1. indicates DINT Attentive layer and Differential Attentive Layer are used together. However, from the writing in Section 2.5, DFPAL refers to Differential Attentive Layer and DINTAL refers to DINT Attentive Layer. Is this a discrepancy?

---

### Official Review · Reviewer_6JFB · 2025-10-29

**Soundness:** 2
**Presentation:** 2
**Contribution:** 2
**Rating:** 4
**Confidence:** 3

**Summary:**

This paper proposes a signal quality classification method based on convolutional neural network (CNN) and combined with differential and integral attention mechanism for the problem of photoplethysmography (PPG) signal quality assessment (SQA).

**Strengths:**

1. The experimental design is well-designed and the dataset is rich. This experiment uses multiple datasets from different devices (Samsung Galaxy Watch 5/6/7 and Galaxy Ring), ensuring the generalizability of the model.
2. Experimental results demonstrate that the proposed method demonstrates good performance in metrics such as accuracy, F1-score, and AUC, particularly demonstrating good adaptability across different hardware platforms.

**Weaknesses:**

1. The introduction fails to clearly demonstrate the specific contributions of the proposed method or its similarities and differences with previous approaches.
2. Differential and integral attention layers appear to be the core contributions of this study, but they are not original.
3. The results section lacks qualitative analysis of the results, such as the addition of interpretability analysis of differential (DIFF) attention and differential integral (DINT) attention. It would be helpful to add discriminant analysis with latent feature visualization.

**Questions:**

Same as weaknesses.

---

### Official Review · Reviewer_hrHY · 2025-10-30

**Soundness:** 3
**Presentation:** 3
**Contribution:** 2
**Rating:** 2
**Confidence:** 5

**Summary:**

This paper proposes a lightweight two-stage framework for PPG signal quality assessment on wearable devices. It first removes obviously corrupted segments using a simple amplitude threshold, then classifies the remaining signals with a compact CNN enhanced by Differential and Differential-Integral attention layers. The model architecture is discovered via Neural Architecture Search (NAS) to ensure efficiency. Experiments on four wearable PPG datasets show high accuracy and generalization, demonstrating suitability for embedded health monitoring applications.

**Strengths:**

The paper adapts differential and integral attention mechanisms to 1D PPG signal quality assessment, the technical quality is solid, with clear architecture design, well-motivated ablations, and consistent results across multiple datasets. The paper is clearly written, presenting equations and experimental procedures in an accessible and organized manner. Its significance lies in demonstrating an efficient and accurate on-device SQA solution that can enhance the reliability of wearable health monitoring systems.

**Weaknesses:**

1. Limited real-world validation: All datasets were collected under resting conditions; no tests were conducted under motion or exercise scenarios, which are crucial for practical wearable applications.

2. Weak novelty in method composition: The DIFF and DINT attention mechanisms are borrowed from prior works; the main novelty lies in their application to PPG, which limits conceptual originality.

3. Insufficient cross-device generalization analysis: Although the study includes four datasets from different Samsung devices, they share similar sensor designs and processing pipelines. Thus, the results mainly show intra-brand consistency rather than true cross-device generalization. Testing on PPG signals from other vendors (e.g., Apple, Fitbit or Huawei) would better validate the model’s robustness and practical applicability.

4. Lack of demographic diversity reporting: The paper does not report participants’ demographic information, such as skin tone, ethnicity, or age distribution. Since optical PPG signals are known to vary with melanin levels, skin thickness, and vascular properties, omitting this information limits the assessment of model generalization across diverse populations and real-world users.

5. Single-expert labeling limits reliability: All PPG segments appear to have been annotated by a single expert. Given the subjective nature of signal quality assessment, relying on one annotator raises concerns about label noise and inter-rater bias. Incorporating multiple experts and reporting inter-rater agreement metrics would provide stronger evidence of labeling reliability and improve the validity of model evaluation.

6. Unclear training protocol: There is an inconsistency between the stated 100-epoch training and the 1000-epoch results table, raising reproducibility concerns.

7. No runtime or deployment benchmarks: Although the method targets embedded devices, there are no latency, FLOPs, or power-consumption measurements to substantiate on-device feasibility.

8. Lack of interpretability or feature visualization: The paper does not provide attention maps or qualitative examples to show what temporal patterns DIFF/DINT actually capture, limiting insight into model behavior.

**Questions:**

Please refer to the weaknesses discussed above.

---

### Official Review · Reviewer_7LK8 · 2025-11-05

**Soundness:** 1
**Presentation:** 3
**Contribution:** 2
**Rating:** 2
**Confidence:** 4

**Summary:**

This paper proposes a two-stage PPG signal-quality classifier for 3-s windows: (i) a fast amplitude-range threshold to discard obvious low-quality segments, followed by (ii) a compact NAS-discovered CNN whose global pooling is replaced by Differential and Differential-Integral attention layers. Experiments are reported on four Samsung datasets (Galaxy Watch 5/6/7 and Galaxy Ring), labeled by a single cardiologist, with subject-wise splits; two-stage models with the new attentions outperform in-paper attention baselines and several classical descriptors.

**Strengths:**

1. Practical framing for on-device use (tiny models + cheap first stage), with a clear architectural description and an ablation on attention scaling init and one- vs two-stage design.

**Weaknesses:**

1. Incomplete and imbalanced baseline coverage. The paper omits several recent, closely related PPG signal-quality / artifact-detection approaches:
a. Chen, Guo, Ding, Hu, & Rudin (2024) — Sparse learned kernels for interpretable and efficient medical time series processing (Nature Machine Intelligence, 6, 1132–1144), doi:10.1038/s42256-024-00898-4.
b. Kasaeyan Naeini, Sarhaddi, Azimi, Liljeberg, Dutt, & Rahmani (2023) — A deep learning–based PPG quality assessment approach for heart rate and heart rate variability (ACM Transactions on Computing for Healthcare, 4(4), Article 24), doi:10.1145/3616019.
2. Four of the five “specific SQA algorithms” cited come from two closely connected publications and lack public code. Two baselines are variants from Lucafo et al., and two are from Garcia Freitas et al.. None of these four provide public code, and Garcia Freitas et al., 2025 is a patent, not peer-reviewed. This concentration raises concerns about diversity of comparisons and reproducibility of baselines.
3. Proprietary, single-vendor datasets limit external validity and reproducibility. All four evaluation sets are non-public and collected solely on Samsung wearables; labels come from a single cardiologist. This prevents third-party reproduction, obscures cross-brand generalization (e.g., Apple/Fitbit/medical PPG), and leaves inter-rater reliability unquantified.

**Questions:**

1. Are there plans to release (even a subset of) GW5/6/7/RING or to benchmark on public, motion-rich sets to support reproducibility?
2. Any “train on GW5/6 → test on GW7/Ring” or cross-brand results to probe generalization beyond a single vendor?
3. Please justify the choice of a 3-second window and report a sensitivity analysis (e.g., 2/3/5 s, with/without overlap).
4. Please report on-device efficiency metrics (latency in ms/inference, energy per inference, and peak RAM/flash) on a representative wearable SoC, not just parameter count.

**Details Of Ethics Concerns:**

The datasets used in this study are not publicly available and lack clear, detailed descriptions of the data collection process.

---

### Note · Authors · 2025-12-08

I have read and agree with the venue's withdrawal policy on behalf of myself and my co-authors.